# Three-Vector-Based Smart Model Predictive Torque Control of Surface-Mounted Permanent Magnet Synchronous Motor Drives for Robotic System Based on Genetic Algorithm

**Shenghui Li** [1,*]**, Li Ma** [2]**, Jingrui Hou** [3]**, Yiqing Ma** [2] **and Rongbo Lai** [2]

[1] School of Electronic and Optical Engineering, Nanjing University of Science and Technology Zijin College, Nanjing 210023, China
[2] School of Electrical and Information Engineering, Jiangsu University, Zhenjiang 212013, China; mali@ujs.edu.cn (L.M.); myq@stmail.ujs.edu.cn (Y.M.); 2222307080@stmail.ujs.edu.cn (R.L.)
[3] School of Engineering, RMIT University, Melbourne, VIC 3001, Australia; s3837655@student.rmit.edu.au
* Correspondence: lishenghuinj@hotmail.com

**Abstract:** Owing to their high performance and high-efficiency controllability, surface-mounted permanent magnet synchronous motors (SPMSMs) have been widely considered for various robotic systems. The conventional three-vector-based model predictive torque control (MPTC) is frequently applied to SPMSMs, while the adjustment of weight factors is difficult. Compared with the five-segment sequence output method, the three-segment sequence output method can effectively reduce the switching frequency. However, the three-segment sequence output method leads to large torque and stator flux ripple. For these issues, a three-vector-based smart MPTC method based on the optimal vector sequence optimized by a genetic algorithm is proposed. Firstly, the reference voltage vector output from the discrete-time sliding mode (DTSM) current controller is utilized to simplify the process of selecting the vectors, and it can enhance the robustness of the SPMSM system. Secondly, an improved cost function is employed to select the optimal vector sequence, aiming to minimize torque and flux ripple. Furthermore, the multi-objective genetic algorithm is leveraged to seek the Pareto solution for weight factors. As a final step, the efficacy of the designed MPTC approach is confirmed through simulations and experiments.

**Keywords:** genetic algorithm; finite control set MPC; optimal vector sequence; model parameter mismatch; SPMSM

## 1. Introduction

In recent decades, SPMSMs have gained considerable attention in robotic systems due to their small size, reliable operation, and high efficiency [1–3]. Generally speaking, among the control strategies for SPMSM drive systems, field-oriented control (FOC) and direct torque control (DTC) are the predominant approaches. Although DTC and FOC methods can obtain high tracking and good steady-state performance, there remains a need for further improvement in dynamic response speed. Consequently, there has been considerable attention paid to model predictive control (MPC) in recent years, since this method can predict system behavior with nonlinearities and multiple constraints [4–7].

Broadly speaking, the MPC approaches can be classified into two main types: continuous control set MPC (CCS-MPC) and finite control set MPC (FCS-MPC). Typically, CCS-MPC necessitates the use of a modulator, while FCS-MPC does not [8]. CCS-MPC generates a continuous control signal to the modulation stage, offering advantages such as rapid dynamics and stable switching frequencies [9]. Nevertheless, it is difficult to be

directly optimized online because of the great computational burden. To take away this problem, the offline calculation algorithm was adopted in [10]. In addition, the motor was driven by discrete switching values of FCS-MPC in [11], which contains several merits such as its quicker dynamics, intuitive concept of design, and straightforward structure. For the motor drive system, in order to obtain a fine dynamic response, FCS-MPC can substitute the conventional proportional–integral (PI) cascade control method [12].

Generally speaking, the FCS-MPC methods are categorized into various types depending upon the different predictor variables [13]. In model predictive current control (MPCC), the predictor variable is predicated on the current signal, meaning that the cost function exclusively accounts for the current signal as the variable, while the weighting factors do not need to be computed. The predictor variables of MPTC are torque and flux, which must be taken into account by the cost function as control variables. Thus, the weighting factor design of the cost function is required. However, there is no systematic and reliable weighting factor design scheme in MPTC, and the weighting factor adjustment in practice requires a large number of experiments [14,15]. For the purpose of simplifying the weighting factor rectification, the Pareto solution of the weighting factors is found using a multi-objective genetic algorithm in the literature [16]. In addition, MPCC exhibits a lower current ripple compared to MPTC, although MPTC demonstrates a reduced torque ripple [17]. Both of them have their own control characteristics.

It should be pointed out that the design of the MPTC is mainly based on one-vector and two-vector control schemes, while neither of them can theoretically eliminate the steady-state error. Consequently, in pursuit of control systems with no steady-state error, the three-vector-based MPTC methods outlined in [18,19] have been put forward to further enhance steady-state error performance. A three-vector-based MPTC of a PMSM for electric vehicles is designed in [18]. To alleviate the complexity and computational load of the MPTC scheme, an improved switching table to facilitate the direct selection of the optimal voltage vector has been devised in [19]. Although the above-mentioned three-vector-based MPTC schemes can significantly enhance the steady-state error, they all utilize five-segment outputs with a high inverter switching frequency. When it comes to high-power applications, a three-segment output is adopted to avoid excessive switching losses, despite the potential for heightened torque and flux ripple due to the reduced switching frequency. In [20], motor performance is improved by opting for the vector sequence of minimum flux. It is noteworthy that sliding mode control is an effective robust control method in dealing with uncertian systems [21–27]. The DTSM method has become more and more popular with the advancement of digital microprocessor technology [28–32]. Due to the possibility of reduced switching gains, the reference voltage vector of the DTSM output can simplify the process of voltage vector selection. Moreover, it should be highlighted that the stability analyses conducted in the continuous systems are not directly applicable to digital control systems. Consequently, the evolution of discrete-time approaches is imperative to ensure the stability of application programs based on micro controllers and facilitate the direct derivation of essential difference equations.

In view of the above analysis, an improved three-vector-based MPTC is designed to enhance the control performance. First, for the purpose of enhancing the robustness of SPMSMs, the SMC is combined with the beat deadbeat predictive torque control approach. The whole voltage vector plane is delineated into six sectors. The DTSM current controller is employed to derive the reference voltage vector, leading to the swift targeting of the target set of voltage vectors based on the sector in which the reference voltage vectors are located. Using the proposed method, a large number of cost function calculations of different voltage vector combinations can be avoided, thus reducing a large amount of the computational burden. Next, through the improved cost function, the voltage vector output

sequence with the minimum torque and stator flux ripple will be identified. Currently, AI technology is widely applied in practical engineering fields [33–38]. Tasks such as parameter tuning can be replaced by AI [39–44]. Consequently, the weight factor's Pareto solution is found using a multi-objective genetic algorithm with steady-state torque ripple, switching frequency, and stator flux ripple as optimization targets.

The organization of this paper is outlined below. Section 2 details the mathematical models for the SPMSM, inverter model, and the conventional FCS-MPTC. The detailed procedure of the designed MPTC method is outlined in Section 3. Comparative simulation results between the designed and conventional MPTC approaches are presented in Section 4. The comparison of experimental results of the proposed MPTC with PI is given in Section 5. Finally, Section 6 will give the conclusions.

## 2. Conventional FCS-MPC of an SPMSM Drive System

### 2.1. SPMSM Mathematical Model

The mathematical formulation of the SPMSM within the $d - q$ axis synchronous rotating frame is shown below:

$$\begin{cases} i_d = \dfrac{1}{L_s}u_d - \dfrac{R_s}{L_s}i_d + \omega_e i_q \\ i_q = \dfrac{1}{L_s}u_q - \dfrac{R_s}{L_s}i_q - \omega_e i_d - \dfrac{\psi_f \omega_e}{L_s} \end{cases} \tag{1}$$

where $\omega_e$ is the electrical angular velocity of the SPMSM, $i_d$ and $i_q$ are the components of the vector representing the direct and quadrature axis currents, $u_d$ and $u_q$ are the $d - q$ axis components of the stator voltages, and $L_s$, $R_s$, and $\psi_f$ are the inductance, resistance, and flux linkage of the stator of the SPMSM, respectively.

By utilizing the forward Euler method on (1), one can represent the discrete-time current model of the SPMSM as

$$\begin{cases} i_d(k+1) = (1 - \dfrac{T_s R_s}{L_s})i_d(k) + T_s \omega_e i_q(k) + \dfrac{T_s}{L_s}u_d(k) \\ i_q(k+1) = (1 - \dfrac{T_s R_s}{L_s})i_q(k) - T_s \omega_e i_d(k) - \dfrac{T_s \omega_e \psi_f}{L_s} + \dfrac{T_s}{L_s}u_q(k), \end{cases} \tag{2}$$

where the symbol $T_s$ represents the time interval between two consecutive samples.

Based on (2), the expression for the flux linkage of the SPMSM in the $d - q$ axis is shown below:

$$\begin{cases} \psi_d(k+1) = T_s\big[u_d(k) - R_s i_d(k) + \omega_e \psi_q(k)\big] + \psi_d(k) \\ \psi_q(k+1) = T_s\big[u_q(k) - R_s i_q(k) - \omega_e \psi_d(k)\big] + \psi_q(k), \end{cases} \tag{3}$$

where $\psi_d$ and $\psi_q$ are the stator flux linkage components of the $d - q$ axis.

For ease of description, the mathematical representation of the SPMSM model can be rewritten as shown below:

$$\begin{cases} x(k+1) = Ax(k) + Bu(k) + W(k) \\ y(k) = Cx(k) \end{cases} \tag{4}$$

where $u$ is the input vectors, $x$ is the state vectors, $y$ is the output vectors, and $W(k)$ represents disturbance term, with

$$A = \begin{bmatrix} 1 - \frac{T_s R_s}{L_s} & T_s \omega_e(k) \\ -T_s \omega_e(k) & 1 - \frac{T_s R_s}{L_s} \end{bmatrix}, C = \begin{bmatrix} 1 & 0 \\ 0 & 1 \end{bmatrix}, \tag{5}$$

$$B = \begin{bmatrix} \frac{T_s}{L_s} & 0 \\ 0 & \frac{T_s}{L_s} \end{bmatrix}, W(k) = \begin{bmatrix} 0 \\ -\frac{T_s \omega_e(k)\psi_f}{L_s} \end{bmatrix}. \tag{6}$$

Here, $x(k) = [i_d \quad i_q]^T$, $u(k) = [u_d \quad u_q]^T$, and $y(k) = [i_d \quad i_q]^T$. $C$ is the identity matrix. Moreover, the disturbance term $W(k)$ contains the electric speed $\omega_e(k)$.

On this basis, the predicted values of the torque and flux of the SPMSM are expressed as shown below:

$$\begin{cases} \psi_s(k+1) = T_s[u_s(k) - R_s i_s(k)] + \psi_s(k) \\ T_e(k+1) = 1.5p[\psi_s(k+1) \otimes i_s(k+1)] \end{cases} \tag{7}$$

with

$$\begin{cases} \psi_s(k+1) = \sqrt{\psi_d(k+1)^2 + \psi_q(k+1)^2} \\ i_s(k+1) = i_s(k) + \dfrac{T_s}{L_s}[u_s(k) - R_s i_s(k) - j\omega_e \psi_s(k)], \end{cases} \tag{8}$$

where $i_s$, $u_s$, and $\psi_s$ are the current, voltage, and flux of the stator, respectively, and $p$ is the number of pole pairs.

### 2.2. Model of Inverter

The SPMSM system in this paper uses a two-level inverter with a classical configuration, as described in [11]. The inverter's initial voltage vectors are derived using the switching function model defined in different reference frames. These voltage vectors are obtained as follows:

$$u = T_P \cdot T_C \cdot u_i \tag{9}$$

$$T_P = \begin{bmatrix} \cos(\theta_e) & \sin(\theta_e) \\ -\sin(\theta_e) & \cos(\theta_e) \end{bmatrix} \tag{10}$$

$$T_C = \frac{2}{3} \begin{bmatrix} 1 & -1/2 & -1/2 \\ 0 & \sqrt{3}/2 & -\sqrt{3}/2 \end{bmatrix}. \tag{11}$$

where $T_P$ and $T_C$ are the Park and Clark transformations, respectively. All possible voltage vector values $u_i(i = 0, 1, 2, \ldots, 7)$ are displayed in Figure 1, in which $\theta_e$ represents the rotor's electrical angle and $U_{dc}$ is the DC-link voltage.

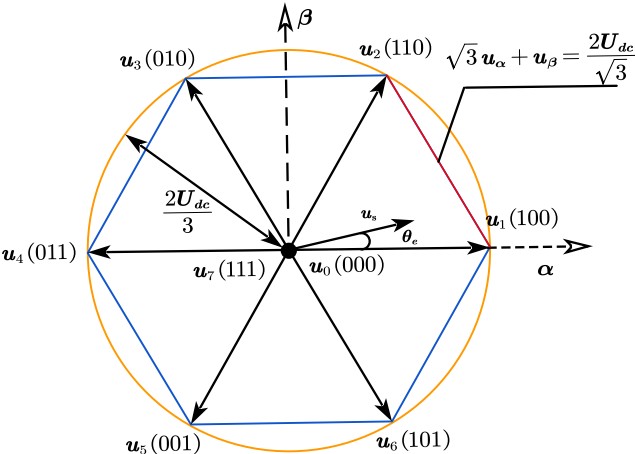

**Figure 1.** The diagram of voltage vectors for a two-level inverter.

### 2.3. Convention FCS-MPTC

For the FCS-MPTC in SPMSM drives, the control objective is that the prediction value should be as close as possible to the flux reference value and torque reference value. Consequently, the formulation of the cost function is concisely articulated as follows:

$$g = |T_e^* - T_e(k+1)| + c_\psi |\psi_s^* - \psi_s(k+1)| + g_{\lim} \tag{12}$$

where $c_\psi$ is the flux weighting factor; $T_e^*$ represents the reference value of torque, which is obtained from the discrepancy between the given speed and the actual speed by PI adjustment; $\psi_s^*$ is the flux reference value, calculated from the maximum torque per ampere; and $g_{lim}$ is a safety constraint to ensure that the system operates within safe limits. $g_{lim}$ is defined as follows:

$$g_{lim} = \begin{cases} 0, & \text{if} \quad \|y(k+1)\|_2 \le i_{smax} \\ \infty, & \text{if} \quad \|y(k+1)\|_2 > i_{smax} \end{cases} \tag{13}$$

where $y(k+1)$ represents the predicted current and $i_{smax}$ is the current limitation. The control scheme of the conventional MPTC strategy is implemented using several building blocks, illustrated in Figure 2.

The optimal future input $u(k)$ applied to the SPMSM is computed based on the minimization of the following cost function $g$:

$$\min g = \min_{u_i}\left\{ \|y^*(k+1) - y(k+1)\|_Q^2 + g_{lim} \right\}$$

$$\text{s.t} \quad x(k+1) = \boldsymbol{A}x(k) + \boldsymbol{B}u(k) + \boldsymbol{W}(k) \tag{14}$$

$$y(k) = \boldsymbol{C}x(k), u_i = \{u_0, u_2, \ldots, u_7\}$$

where $y = [i_d \quad i_q]^{\mathrm{T}}$ represents the predicted value, and $y^* = [i_d^* \quad i_q^*]^{\mathrm{T}}$ is the reference current.

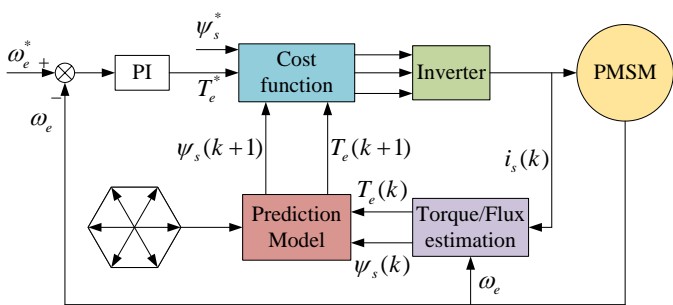

**Figure 2.** Block schematic of the conventional MPTC strategy.

## 3. Proposed Control Method

### 3.1. DTSM Current Controller

A DTSM current controller is proposed in this subsection to reduce three-vector-based MPTC computation and enhance the anti-interference ability of the control system.

We, respectively, select the following sliding surface and the exponent reaching law as $s = c \int (y^* - y)$ and $\dot{s} = -\frac{|s|}{2}\text{sgn}(s) - \eta s$, where $c$ is the positive integral parameter and $s = [s_d \quad s_q]^T$, $y = [i_d \quad i_q]^T$, and $y^* = [i_d^* \quad i_q^*]^T$. Moreover, $-\frac{|s|}{2}\text{sgn}(s)$ represents the adaptive reaching term, and $\eta s$ is the positive index reaching term. In the simulation and experiment, we select the parameter values as $c = 0.5$ and $\eta = 50$.

To design the DTSM current controller, we define the $\delta$ operator as

$$\delta f(k) = \frac{f(k+1) - f(k)}{T_s} \tag{15}$$

where $f(k)$ represents a generic function at the $k$th period, which can be the sliding surface or currents.

Using (15), we can obtain

$$\begin{aligned} \delta s(k+1) &= \delta s(k) + T_s \delta^2 s(k) \\ &= \delta s(k) - c[y(k+1) - y(k)] \\ &= -\frac{|s(k+1)|}{2}\text{sgn}[s(k+1)] - \eta[s(k+1)] \end{aligned} \tag{16}$$

where $\delta s(k)$ is the discrete derivative of the sliding surface at the $k$th period; $\delta^2 s(k)$ is the discrete second derivative of the sliding surface at the $k$th period

According to (4) and (16), we can obtain the reference voltage(4)

$$
\begin{aligned}
u^*(k) = \frac{\boldsymbol{B}^{-1}}{c} \Big( & \delta s(k) + \frac{|s(k+1)|}{2} \mathrm{sgn}[s(k+1)] \\
& + \eta[s(k+1)] + c(\boldsymbol{I} - \boldsymbol{A})x(k) - c\boldsymbol{W}(k) \Big)
\end{aligned}
\tag{17}
$$

Here, the symbol $\boldsymbol{I}$ is utilized to signify the identity matrix.

Next, we will show the stability by constructing a finite-time Lyapunov function as shown below:

$$
V(k) = \frac{1}{2}s^2(k)
\tag{18}
$$

From (15) and (18), it is derived that

$$
\begin{aligned}
\delta V(k) &= s(k)\delta s(k) \\
&= \left\{ -\frac{|s(k)|}{2}\mathrm{sgn}[s(k)] - \eta[s(k)] \right\} s(k) \\
&= -(0.5 + \eta)s^2(k) < 0
\end{aligned}
\tag{19}
$$

Through the above proof, it is shown that the existence and arrival conditions for the designed discrete sliding mode are satisfied. In light of the analysis above, taking the reference voltage of the quadrature axis as an example, Figure 3 illustrates the block diagram of the DTSM current controller.

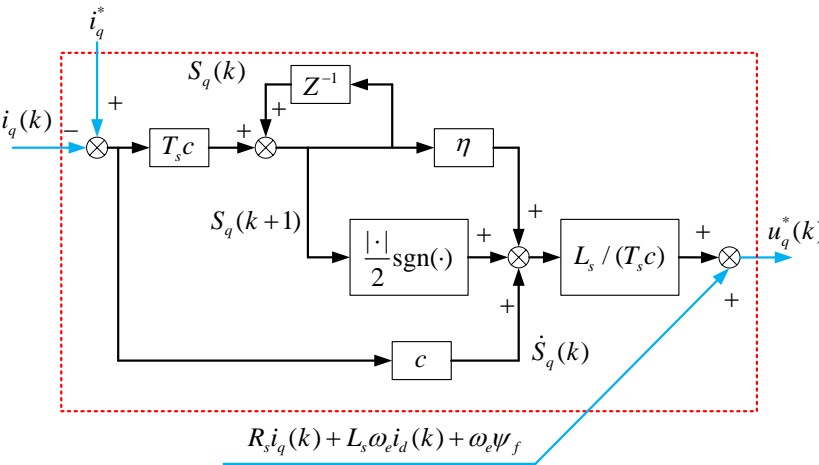

**Figure 3.** System block diagram of DTSM current controller.

Subsequently, the reference voltage's phase angle $\theta^*$ is determined as follows:

$$
\theta^* = \arctan\left[\frac{u_\beta^*}{u_\alpha^*}\right]
\tag{20}
$$

The conventional three-vector-based MPTC needs to select three vectors from $u_i$ for each control period, necessitating a cost function calculation for each potential combination, which has a huge computational burden. To avoid wasting the control system's computational capacity, the number of comparisons needs to be reduced. Figure 4 shows that when the voltage reference is located at a known sector, the target voltage vectors will be determined to be a combination of two neighboring vectors and a single zero vector. The association between the sector and the target vectors is delineated in Table 1.

**Table 1.** Target vector lookup table.

| Sector | The Target Vectors |
|---|---|
| I | $u_1, u_2, u_0$ or $u_7$ |
| II | $u_2, u_3, u_0$ or $u_7$ |
| III | $u_3, u_4, u_0$ or $u_7$ |
| IV | $u_4, u_5, u_0$ or $u_7$ |
| V | $u_5, u_6, u_0$ or $u_7$ |
| VI | $u_1, u_6, u_0$ or $u_7$ |

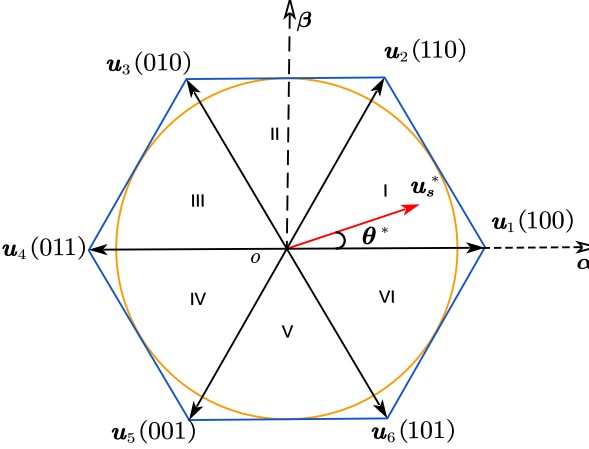

**Figure 4.** Voltage space vector diagram.

The relationship between the durations of the target voltage vectors can be expressed as follows:

$$u_\alpha^* T_s = u_{1\alpha} T_1 + u_{2\alpha} T_2 \tag{21}$$

$$u_\beta^* T_s = u_{1\beta} T_1 + u_{2\beta} T_2 \tag{22}$$

where $T_1$ and $T_2$ are, respectively, the durations of the two nonzero voltage vectors; $u_{1a}$ and $u_{2a}$ and $u_{1\beta}$ and $u_{2\beta}$ are the $\alpha$ and $\beta$ axis voltage components of the two nonzero vectors, respectively. Consequently, $T_1$, $T_2$, and $T_3$ can be obtained as

$$T_1 = \frac{u_\beta^* u_{2a} - u_\alpha^* u_{2\beta}}{u_{1\beta} u_{2\alpha} - u_{1\alpha} u_{2\beta}} T_s \tag{23}$$

$$T_2 = \frac{u_\beta^* u_{1a} - u_\alpha^* u_{1\beta}}{u_{2\beta} u_{1\alpha} - u_{2\alpha} u_{1\beta}} T_s \tag{24}$$

$$T_3 = T_s - T_2 - T_1 \tag{25}$$

where $T_3$ is the duration of $u_{0,7}$.

### 3.2. Predictive Control with the Optimal Vector Sequence

In this section, the three-segment sequence output method is selected. It implies that each voltage vector can only be applied once in a control period, which can greatly reduce the switching frequency compared with the five-segment sequence output method. And by finding the optimal sequence of voltage vectors, the ripple growth resulting from the reduced switching frequency can be reduced.

Taking the first sector as an example, adopting the principle that the two-level inverter switch only changes once and choosing the zero vector reasonably, there are four kinds of three-segment sorting ( Sequence *A*: $u_1$ $u_2$ $u_7$; Sequence *B*: $u_2$ $u_1$ $u_0$; Sequence *C*: $u_0$ $u_1$

$u_2$; and Sequence $D$: $u_7\ u_2\ u_1$). Table 2 exhibits the possible voltage vector sequences for each sector.

**Table 2.** Possible voltage vector sequences for each sector.

| Sector | *A* | *B* | *C* | *D* |
|:---:|:---:|:---:|:---:|:---:|
| I | $u_1\ u_2\ u_7$ | $u_2\ u_1\ u_0$ | $u_0\ u_1\ u_2$ | $u_7\ u_2\ u_1$ |
| II | $u_3\ u_2\ u_7$ | $u_2\ u_3\ u_0$ | $u_0\ u_3\ u_2$ | $u_7\ u_2\ u_3$ |
| III | $u_3\ u_4\ u_7$ | $u_4\ u_3\ u_0$ | $u_0\ u_3\ u_4$ | $u_7\ u_4\ u_3$ |
| IV | $u_5\ u_4\ u_7$ | $u_4\ u_5\ u_0$ | $u_0\ u_5\ u_4$ | $u_7\ u_4\ u_5$ |
| V | $u_5\ u_6\ u_7$ | $u_6\ u_5\ u_0$ | $u_0\ u_5\ u_6$ | $u_7\ u_6\ u_5$ |
| VI | $u_1\ u_6\ u_7$ | $u_6\ u_1\ u_0$ | $u_0\ u_1\ u_6$ | $u_7\ u_6\ u_1$ |

Diverse output sequences of voltage vectors can induce varying degrees of current fluctuation [45]. Such variations in current are directly linked to discrepancies in torque and flux. Taking Figure 5 as an example, in a control period, the error accumulation of the flux will be different due to the different sequences of voltage vector action. To obtain the output sequence of the minimum average ripple within a single control period, the accumulated torque and flux errors after each vector switch should be considered as constraints in the cost function. Furthermore, the switching between adjacent control periods should be taken into account as a constraint to minimize the number of inverter switch changes.

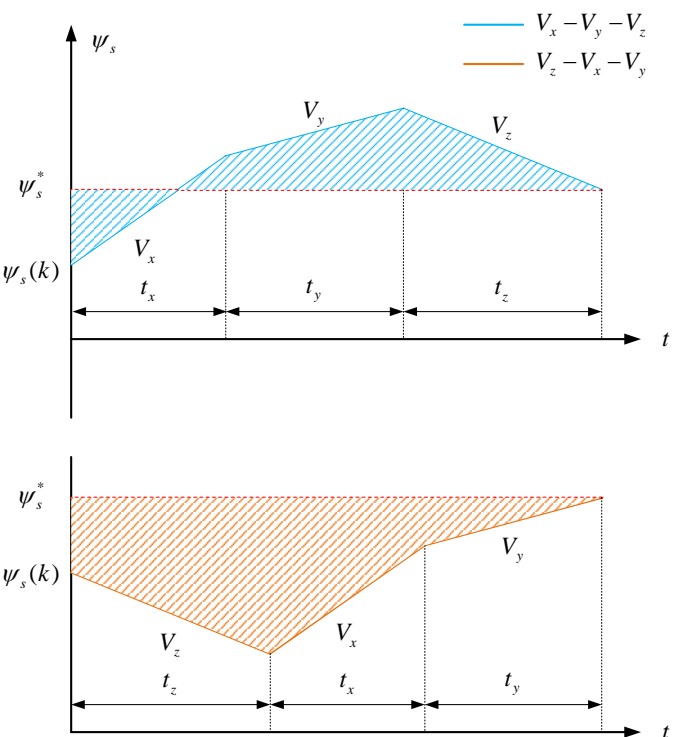

**Figure 5.** Stator flux fluctuation of two different voltage vector sequences.

The current after each switching of the voltage vector can be constructed using the following equation:

$$\begin{cases} i_d^{(n+1)} = \left(1 - \frac{R_s T^{(n)}}{L_d}\right) i_d^{(n)} + \frac{T^{(n)}}{L_d} u_d^{(n)} + \omega_e(k) T^{(n)} i_q^{(n)} \\ i_q^{(n+1)} = \left(1 - \frac{R_s T^{(n)}}{L_q}\right) i_q^{(n)} + \frac{T^{(n)}}{L_q} u_q^{(n)} - \omega_e(k) T^{(n)} i_d^{(n)} - \frac{T^{(n)}}{L_q} \omega_e(k) \psi_f \end{cases} \tag{26}$$

where $n$ = 1, 2, and 3; $i_d^{(1)} = i_d(k)$ and $i_q^{(1)} = i_q(k)$; and $u_d^{(n)}$ and $u_q^{(n)}$ are the $d-q$ axis components of the $n$th applied voltage vector. $T^{(n)}$ is the duration of the $n$th applied voltage vector.

The accumulation of torque error during a control period can be articulated as follows:

$$g_t = \sum_{n=1}^{3} |T_e^* - T_e^n| T^{(n)} \tag{27}$$

where $T_e^{(n)} = 1.5 p \psi_f i_q^{(n)}$ and $T_e^* = 1.5 p \psi_f i_q^*$. And $i_q^*$ can be obtained from the PI controller of the velocity loop.

The stator flux after each switching of the voltage vector can be calculated as follows:

$$\psi_s^{(n)} = \sqrt{\left(L_d i_d^{(n)} + \psi_f\right)^2 + \left(L_q i_q^{(n)}\right)^2} \tag{28}$$

According to (28), we can obtain the error accumulation of the stator flux in one control period as follows:

$$g_\psi = \sum_{n=1}^{3} \left|\psi_s^* - \psi_s^{(n)}\right| T^{(n)} \tag{29}$$

where $\psi_s^* = \sqrt{\left(\psi_f\right)^2 + \left(\frac{L_q T_e^*}{1.5 p \psi_f}\right)^2}$.

The change in switching state between adjacent control periods can be calculated as in (30).

$$\begin{aligned} g_{sw} = 2(&|S_a^{(1)}(k+1) - S_a^{(3)}(k)| + |S_b^{(1)}(k+1) - S_b^{(3)}(k)| \\ &+ |S_c^{(1)}(k+1) - S_c^{(3)}(k)|) \end{aligned} \tag{30}$$

where $S_a^{(3)}(k)$, $S_b^{(3)}(k)$, and $S_c^{(3)}(k)$ are the inverter switching states corresponding to the last voltage vector of the $k$th control period. $S_a^{(1)}(k+1)$, $S_b^{(1)}(k+1)$, and $S_c^{(1)}(k+1)$ are the inverter switching states corresponding to the first voltage vector of the $(k+1)$th control period.

The cost function used in this article is composed of the addition of (27), (29), and (30):

$$G = g_t + k_1 g_\psi + k_2 g_{sw} \tag{31}$$

where $k_1$ is the flux weighting factor and $k_2$ is the switching frequency weighting factor.

### 3.3. MPTC Multi-Objective Optimization

The weighting factors of the MPTC cost function are always complicated and difficult to adjust. Thankfully, as artificial intelligence advances, more sophisticated algorithms are becoming available for use in tuning weight variables. The genetic algorithm (GA) converges more slowly and is more computationally intensive due to the need to maintain a population and perform multiple iterations. However, it is suitable for global exploration of complex problems like SPMSM multi-objective optimization and is less likely to fall into local optima. Considering that our proposed method only requires offline computation, the effectiveness of multi-objective optimization is the most important factor, rather than convergence speed and computational load. Therefore, the Non-dominated Sorting Genetic Algorithm II (NSGA-II) will be used to find a set of Pareto optimal solutions to resolve the conflicting relationships among flux ripple, torque ripple, and switching frequency in this article. The steps of NSGA-II can be shown as follows [46].

(1) An initial population of $N_1$ individuals is generated.
(2) $N_2$ new individuals are generated from the initial population by crossover.
(3) In order to prevent a local optimum, $N_3$ new individuals are then generated by random mutation.
(4) All the individuals from the first three steps are sorted in a non-dominated order to pick the current Pareto solution set. If the current Pareto solution set is larger than $N_1$, the crowding distance of each optimal solution is calculated, and then some of the

solutions with a small crowding distance are eliminated. The remaining individuals are used as the initial population for the next cycle.

(5)　Until it reaches the criteria for stopping, the algorithm will continue.

Each individual has two genes, which are the flux weighting factor $k_1$ and the switching frequency weighting factor $k_2$. $k_1$ is set in the range of 0.1 to 80 while $k_2$ is set in the range of 0 to 0.00001. The code for the NSGA-II will be executed in the Global Optimization Toolbox that comes with matlab. The maximum torque ripple $T_{rip}$, maximum flux ripple $\psi_{rip}$, and switching frequency $f_s$ at steady state are selected as the optimization objectives [16]. The algorithm settings are shown in Table 3.

**Table 3.** NSGA-II parameter table.

| Parameter | Value |
|---|---|
| Initial population size | 30 |
| Number of generations | 20 |
| Crossover | 0.8 |
| Pareto fraction | 0.32 |
| Selection | Tournament |

Through multiple iterations, the Pareto front is obtained as shown in Figure 6a, and the solutions corresponding to it are depicted in Figure 6b. Considering the importance of torque and avoiding a high switching frequency, we choose the flux weighting factor $k_1 = 65.43$, and the switching frequency weighting factor $k_2 = 7.77 \times 10^{-6}$, respectively.

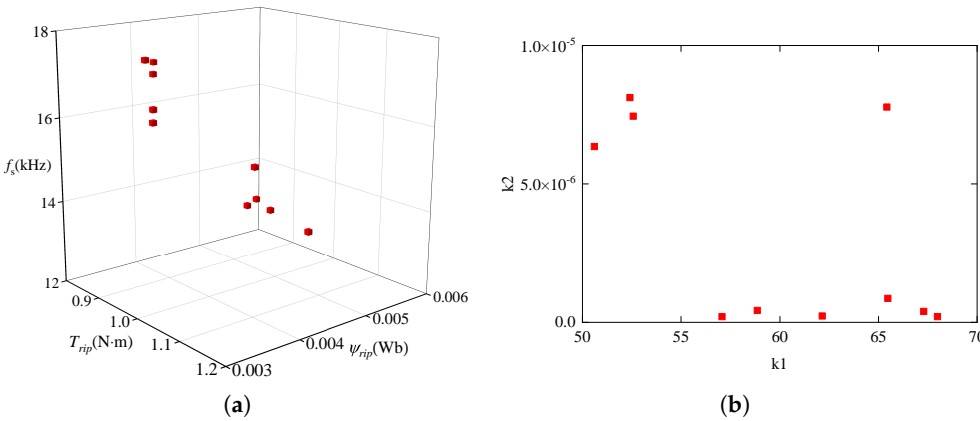

(a)　　　　　　　　　　　　　　　(b)

**Figure 6.** (**a**) Pareto front. (**b**) The corresponding Pareto solutions for a steady-state simulated condition of 500 r/min and 3 N·m load.

The block diagram of the proposed MPTC strategy is shown in Figure 7. To facilitate a better understanding of the proposed MPC method, the detailed implementation steps are summarized as follows:

(1)　According to (17), we can obtain the reference voltage $u^*(k)$.

(2)　The phase angle $\theta^*$ can be obtained by (20).

(3)　According to Table 1, three target vectors are selected at the $k$th period.

(4)　The durations of the three vectors are obtained by (23)–(25).

(5)　The sequence with the smallest output ripple among the four sequences is found by computing (31). And the appropriate weighting factors have been identified in advance by NSGA-II algorithms.

(6)　The corresponding optimal switching state $[S_a \; S_b \; S_c]$ is applied in the drive system.

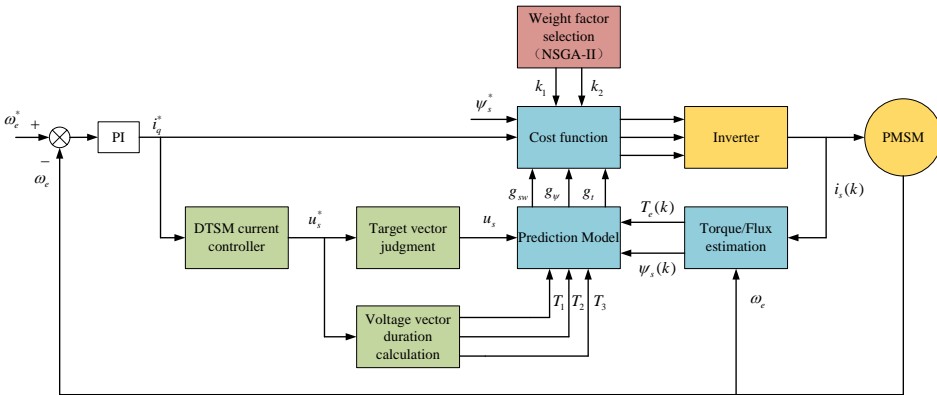

**Figure 7.** Block schematic of the proposed MPTC strategy.

## 4. Simulation Results

The simulation models of the conventional one-vector-based MPTC (Method 1) and the proposed MPTC (Method 2) are constructed in MATLAB Simulink in comparison to evaluate the efficacy of the proposed method.

We choose the flux weight factor $k_\psi = T_N/\psi_{sN}$ in the conventional one-vector-based MPTC cost function, where $T_N$ is the amplitude of the rated torque and $\psi_{sN}$ is the amplitude of the rated stator flux [17]. And both methods maintain the same parameters for the PI controller of the velocity loop.

The specific parameters of the SPMSM are listed in Table 4. The sampling frequency used for all two methods in this paper is set to 20 kHz. Additionally, the DC-link voltage $U_{dc}$ is set to 220 V.

**Table 4.** Specification of the SPMSM.

| Parameter | Description | Value |
|-----------|-------------|-------|
| $P_N$ (kW) | Rated power | 1.5 |
| $T_N$ (N·m) | Rated torque | 10 |
| $p$ | Number of poles pairs | 4 |
| $R_s$ (Ω) | Stator resistance | 1.5 |
| $L_s$ (mH) | Stator inductance | 4.37 |
| $\psi_f$ (Wb) | Rotor magnet flux linkage | 0.142 |
| $J$ (kg·m$^2$) | Rotational inertia | 0.00194 |

The simulation results of Method 1 are compared with Method 2. From Figures 8 and 9, the system steady-state performance under Methods 1–2 is compared under the same conditions of 500 r/min and 3 N·m load. The stator flux can be obtained from (7). As shown in Figure 8, Method 1 has a larger torque ripple and stator flux ripple than Method 2. Meanwhile, Figure 9 illustrates that Method 2 has a lower direct axis current and quadrature axis current ripple than Method 1, which implies that there is an extreme improvement in the current quality of Method 2 with respect to Method 1.

The current prediction errors are influenced by the discrepancies in the values of $L_s$ used in the prediction model, as mentioned in [47]. Based on the parameter sensitivity analysis results in [48], it is evident that the variations in the resistance parameters have a negligible effect on the current prediction values. In contrast, the mismatches in the inductance parameter significantly affect the current prediction error.

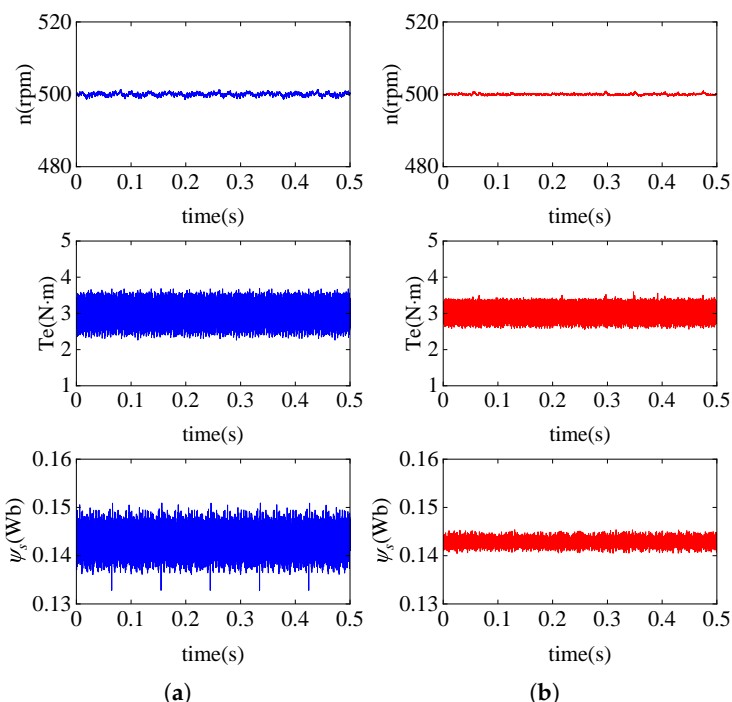

**(a)**        **(b)**

**Figure 8.** Simulation waveforms of speed, torque, and stator flux, with nominal motor parameters at 500 r/min and 3 N·m load steady-state conditions. (**a**) Method 1. (**b**) Method 2.

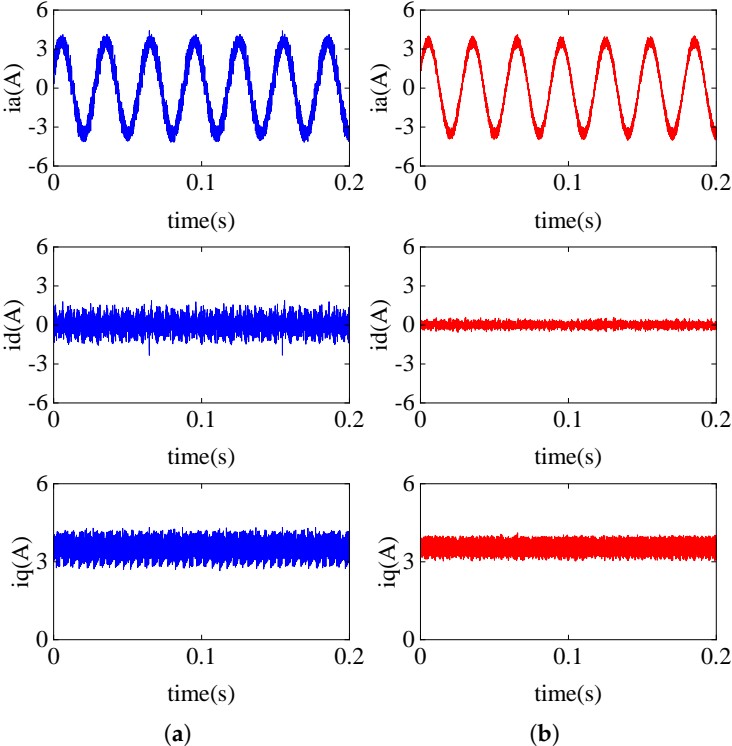

**(a)**        **(b)**

**Figure 9.** Simulation waveforms of A-phase current and *dq* axis current with nominal motor parameters at 500 r/min and 3 N·m load steady-state conditions. (**a**) Method 1. (**b**) Method 2.

To test the feasibility of Method 2 in mitigating parameter mismatches, the inductance parameters of $4L_s$ and $0.25L_s$ are selected. The performance of Methods 1–2 under $4L_s$ with a torque of 3 N·m at a speed of 500 r/min is depicted in Figures 10 and 11. According to Figure 10, the ripple of the torque grows for both methods, and both A-phase currents are distorted as a result. Nevertheless, the overall performance under Method 2 is still superior

to that under Method 1. Figure 11 reveals that the speed ripple of Method 1 grows up to 12.9 rpm, and the A-phase current undergoes a serious deterioration. On the contrary, Method 2 has a strong anti-disturbance capability due to the use of the DTSM current controller. Therefore, the current distortion of Method 2 can be kept in a small range, and the speed and torque ripples do not grow too much.

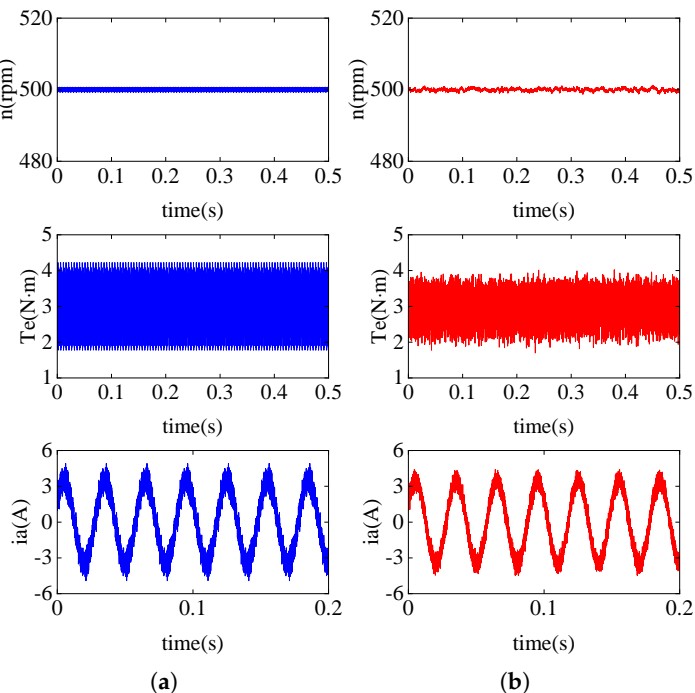

**Figure 10.** Simulation waveforms of speed, torque, and A-phase current with $4L_s$ at 500 r/min and 3 N·m load steady-state conditions. (**a**) Method 1. (**b**) Method 2.

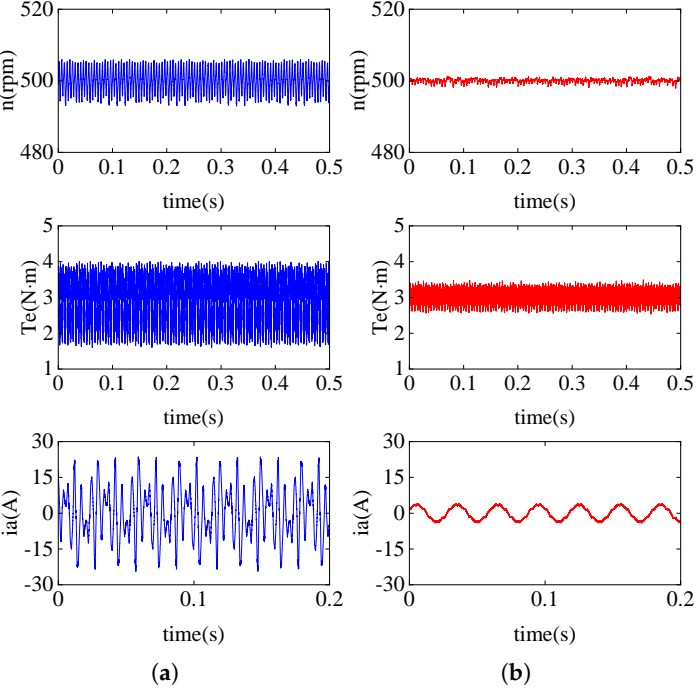

**Figure 11.** Simulation waveforms of speed, torque, and A-phase current with $0.25L_s$ at 500 r/min and 3 N·m load steady-state conditions. (**a**) Method 1. (**b**) Method 2.

To further evaluate the performance differences between the two methods, a new criterion is introduced for quantifying the ripple in the speed or torque, as presented in [49]. The variable ripple can be calculated as

$$x^{rip} = \sqrt{\frac{1}{N}\sum_{i=1}^{N}\|x(i) - x^*\|_2} \tag{32}$$

where $x^{rip}$ is the mean ripple, $x$ represents the sampled value, $N$ is the sample number, and $x^*$ is the reference data.

The speed ripple, torque ripple, and THD of Methods 1–2 are shown in Figure 12. It can be confirmed by comparisons that Method 1 is highly susceptible to the variations in the inductance parameter values. Particularly, when $L = 0.25L_s$, the THD of Method 1 is as high as 317.99%, while the THD of Method 2 is still only 10.82%. This shows that Method 2 is not affected by discrepancies in the inductance parameter values, and it can effectively mitigate the disturbances caused by inductance mismatch.

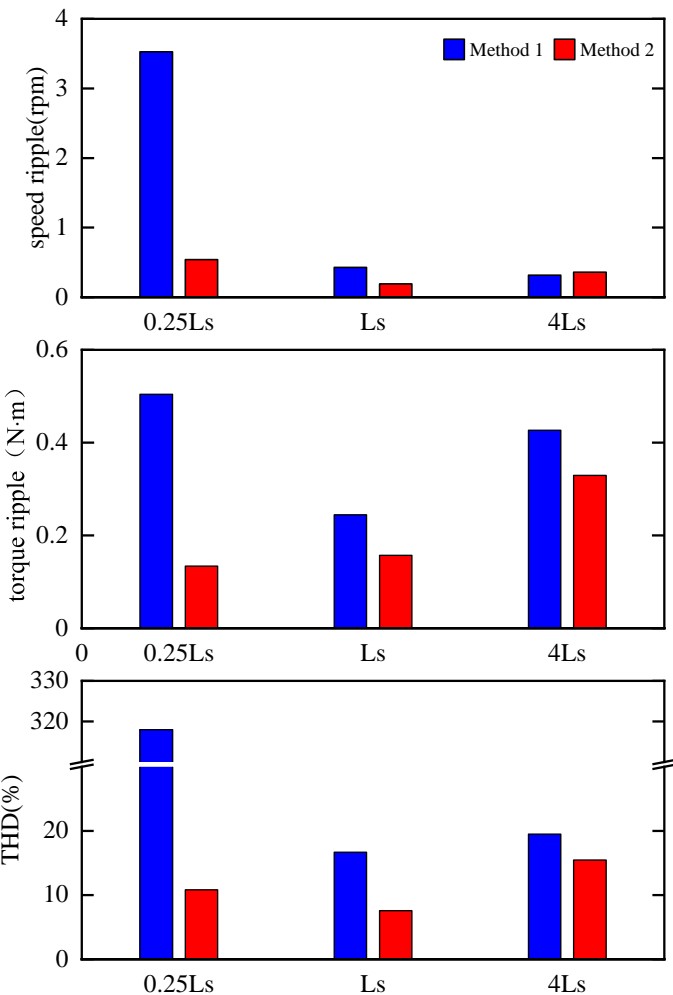

**Figure 12.** Comparison of the speed ripple, torque ripple, and THD of the A-phase stator current with different stator inductances of the three methods.

In order to exhibit the dynamic performance of the proposed Method 2, the reference torque is suddenly increased from 3 N·m to 5 N·m at 500 r/min. It is evident that the proposed Method 2 results in low ripple in both torque and flux linkage according to Figures 13 and 14. The proposed Method 2 provides consistent superior performance across different loading scenarios.

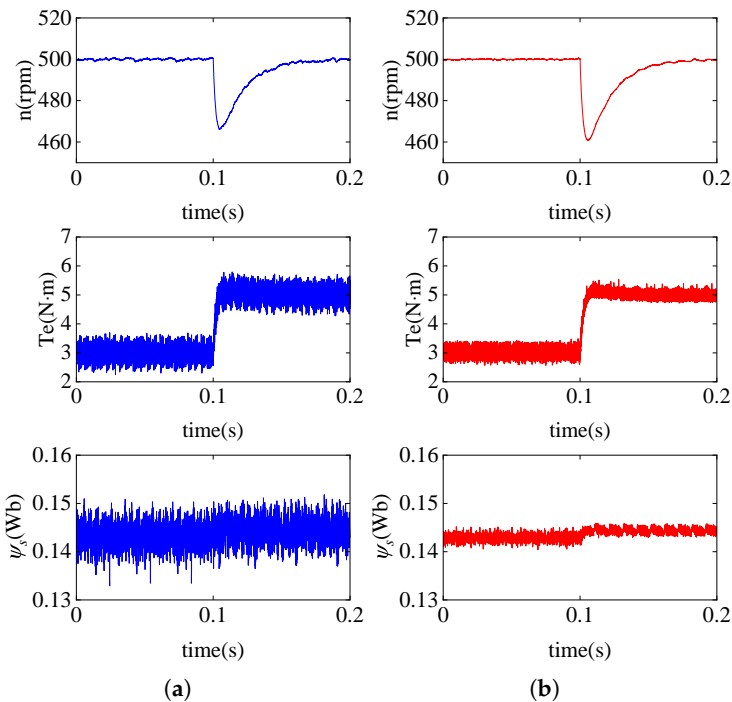

**Figure 13.** Simulation waveforms of speed, torque, and stator flux for a sudden increase in torque from 3 N·m to 5 N·m at 500 r/min. (**a**) Method 1. (**b**) Method 2.

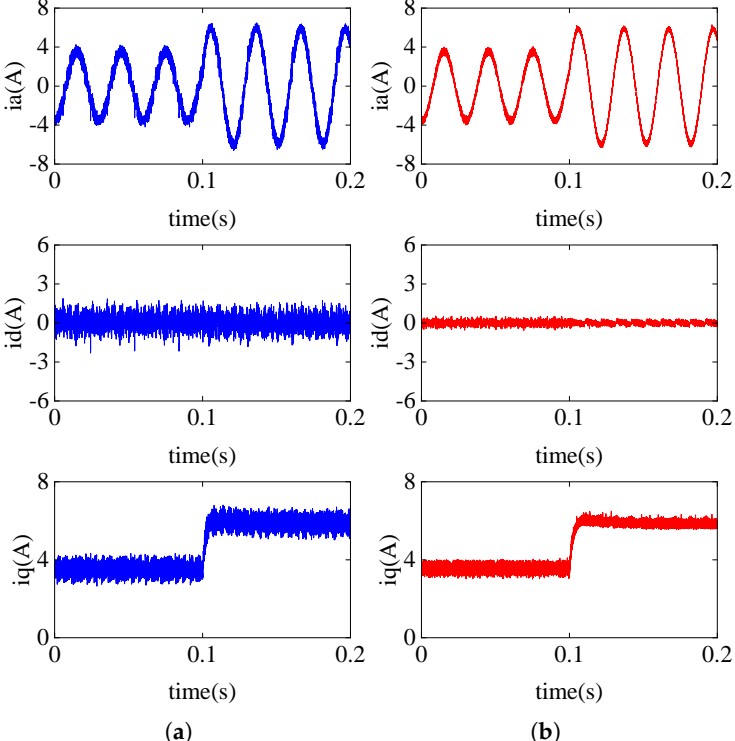

**Figure 14.** Simulation waveforms of A-phase current and *dq* axis current for a sudden increase in torque from 3 N·m to 5 N·m at 500 r/min. (**a**) Method 1. (**b**) Method 2.

## 5. Experimental Results

To evaluate the effectiveness of the MPC method proposed, an experimental platform for controlling an SPMSM has been established. The main control module of the control system selected is the RTU-BOX205. The experimental platform shown in Figure 15 is used to compare the proposed MPTC and the PI controller. And the parameters of the SPMSM

can be obtained from Table 4. The control frequency of the controller for both methods is set to 10 kHz. The flux weighting factor $k_1$ and the switching frequency weighting factor $k_2$ of the proposed MPTC in the experiment are selected as in the simulation. Due to the limitations of the experimental platform, only the cost functions of Sequence A and Sequence C are evaluated in the experiment.

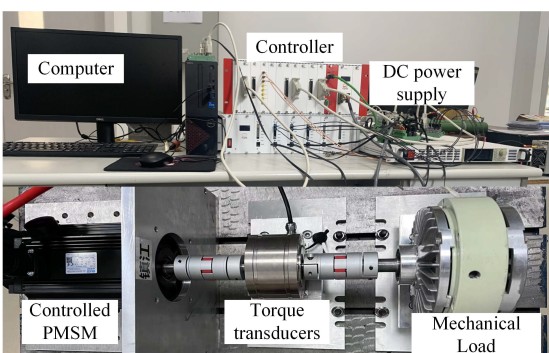

**Figure 15.** Experimental setup of 2-level inverter SPMSM drive.

In Figure 16, the standard deviation of speed ripple is 1.129 r/min for PI and 0.573 r/min for the proposed MPTC. Although both methods employ PI controllers for the speed loops, the speed ripple of the MPTC in Figure 16 is smaller due to the superior current control capability of the proposed MPTC.

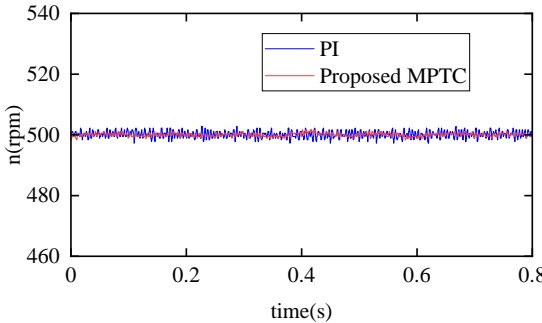

**Figure 16.** Experiment waveforms of speed at 500 r/min steady-state condition.

The use of PI as the current controller requires a seven-segment output using the SVPWM technique, which means that its inverter switching frequency is much higher than that of the proposed MPTC with a three-segment output method. However, the torque ripple of the proposed MPTC is lower than that of the PI, as can be seen in Figure 17. This is due to the fact that the MPTC proposed in this paper is able to evaluate the voltage vector output sequence with the smallest torque ripple in a period, thereby reducing the overall torque ripple during operation.

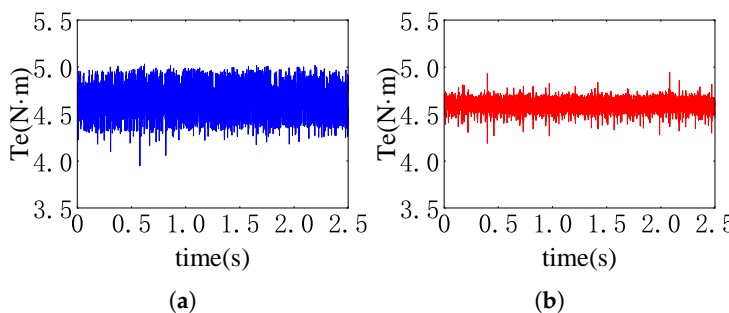

**Figure 17.** Experiment waveforms of torque at 500 r/min steady-state condition. (**a**) PI. (**b**) Proposed MPTC.

The current quality of the SPMSM directly affects its operational performance. In order to visualize the improvement of the proposed MPTC, the A-phase current and its FFT analysis for both the PI and the proposed MPTC are given in Figures 18 and 19. The THD of the PI is 5.7%, while the THD of the proposed MPTC is only 2.37%. The fifth and seventh harmonics of the A-phase current of the PI are significantly higher than those of the proposed MPTC.

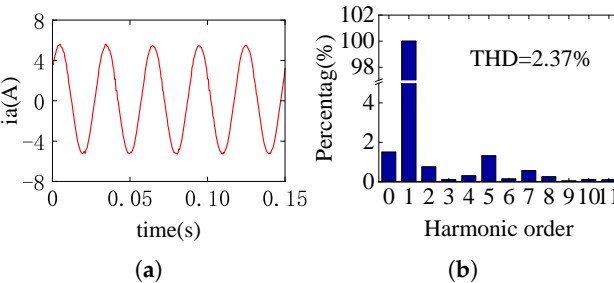

**Figure 18.** Experiment waveforms of proposed MPTC at 500 r/min steady-state condition. (**a**) A-phase current. (**b**) FFT analysis of A-phase current.

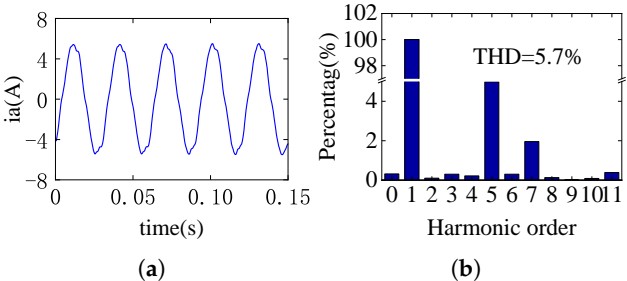

**Figure 19.** Experiment waveforms of PI at 500 r/min steady-state condition. (**a**) A-phase current. (**b**) FFT analysis of A-phase current.

## 6. Conclusions

An enhanced three-vector-based MPTC for an SPMSM is developed in this article. The method leverages the DTSM algorithm to select the target voltage vector. And the weighting factors of the cost function are determined by NSGA-II. Finally, the method employs an improved cost function to find the voltage vector sequence of the minimum average ripple. The analysis of simulation and experimental results can demonstrate that the proposed method ensures the stable operation of the SPMSM and exhibits excellent steady-state performance and enhanced anti-disturbance capabilities, even in the presence of SPMSM parameter mismatches. Unfortunately, although NSGA-II has been used to design weight factors for low switching frequencies, the number of switching changes of the inverter in a control period is higher than that of a conventional one-vector-based MPTC, because three different voltage vectors are continuously used in one control period.

**Author Contributions:** Conceptualization, S.L. and L.M.; methodology, S.L.; software, R.L.; validation, J.H. and Y.M.; writing—original draft preparation, S.L.; writing—review and editing, L.M.; supervision, J.H.; funding acquisition, L.M. All authors have read and agreed to the published version of the manuscript.

**Funding:** This paper is supported by the Nanjing University of Science and Technology Zijin College "High-End Talent Aggregation" Research Initiation Project under grant number 2024ZK0001 and Future Network Scientific Research Fund Project under grant number FNSRFP-2021-YB-53.

**Data Availability Statement:** Data are contained within the article.

**Conflicts of Interest:** The authors declare no conflicts of interest.

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
