# Peer review of "Three-Vector-Based Smart Model Predictive Torque Control of Surface-Mounted Permanent Magnet Synchronous Motor Drives for Robotic System Based on Genetic Algorithm"

_actuators, doi:10.3390/act14030149_

Round 1
Reviewer 1 Report
Comments and Suggestions for Authors
I suggest the authors consider the following points:
1) Given that motor control strategies often behave differently in real-world applications, an experimental prototype or hardware-in-the-loop (HIL) test can improve the study.
2) Since robotic systems require real-time execution, it would be useful to discuss computation time, memory usage, and feasibility for embedded implementation.
3) The study employs genetic algorithms (GA) but doesn’t compare them to other AI-based optimization techniques like particle swarm optimization (PSO) or reinforcement learning (RL).
4) Consider adding a brief discussion of why GA was chosen over other AI techniques and how it compares in terms of convergence speed and computational cost.
5) While the paper discusses the robustness of DTSM, it does not investigate long-term parameter drift effects.
6) A sensitivity analysis on motor inductance, resistance variations, and external load disturbances should add more credibility to the approach.
Author Response
Responses to Reviewer 1
Comment 1. Given that motor control strategies often behave differently in real-world applications, an experimental prototype or hardware-in-the-loop (HIL) test can improve the study.
Response. Thanks for your comment.The method proposed in this paper can output four different sequences of vectors in arbitrary orders according to the cost function within one control period. However, in practice, outputting in the order of Sequence A or Sequence C requires the PWM carrier to be positive triangular, while outputting in the order of Sequence B or Sequence D requires the PWM carrier to be inverse triangular. This means that the controller needs to be able to change the PWM carrier type in real time according to the algorithm. Unfortunately, our experimental platform lacks this capability. However, we have supplemented our experiments with alternative optimal sequences for Sequences A and C to demonstrate the superiority of our proposed algorithm without changing the PWM carrier type. Please see the highlighted parts on Page 16-18 for details.
Comment 2. Since robotic systems require real-time execution, it would be useful to discuss computation time, memory usage, and feasibility for embedded implementation.
Response. Thank you for your comment. We verified the embedded feasibility of the algorithms presented in this paper using an experimental platform containing a TMS320C28346 digital signal processor (DSP). Under the condition that a control period is 100 μs, the computation time of the MPTC is 34.11 μs. Therefore, there is no issue with excessive computation time or memory usage.
Comment 3. The study employs genetic algorithms (GA) but doesn’t compare them to other AI-based optimization techniques like particle swarm optimization (PSO) or reinforcement learning (RL).
Response. Thanks for your comment. The objectives optimized by our proposed method include torque ripple, flux ripple, and inverter switching frequency.The mathematical model of SPMSM is complex, and these three objectives are coupled with each other (e.g., lowering the inverter switching frequency may increase current harmonics due to decreased PWM resolution, which exacerbates torque and flux ripple). Therefore, it is necessary to use intelligent algorithms to perform multi-objective optimization. The particle swarm optimization(PSO) has a lower computational cost and faster convergence speed, but it lacks a variation mechanism and is prone to falling into a local optimum. Reinforcement learning (RL) has a higher computational cost, and its convergence speed is significantly affected by the complexity of the environment. Additionally, an improperly designed reward function can easily lead to policy bias, making it more suitable for scenarios that require online learning or dynamic adjustment. The Genetic Algorithm (GA) converges more slowly and is more computationally intensive due to the need to maintain a population and perform multiple iterations. However, it is suitable for global exploration of complex problems like SPMSM multi-objective optimization and is less likely to fall into local optima. Considering that our proposed method only requires offline computation, the effectiveness of multi-objective optimization is the most important factor, rather than convergence speed and computational load. Therefore, we choose the Genetic Algorithm.
Comment 4. Consider adding a brief discussion of why GA was chosen over other AI techniques and how it compares in terms of convergence speed and computational cost.
Response. Thank you for your comments. We have added a brief discussion about the convergence speed, computational cost, and why GA was chosen over other AI techniques. Please see the highlighted parts on Page 10 for details.
Comment 5. While the paper discusses the robustness of DTSM, it does not investigate long-term parameter drift effects.
Response. Thanks for your comment. The issue you raised about the lack of research on the impact of long-term parameter drift is indeed very important, and we fully agree with your point. However, the inductance parameters of SPMSM typically vary within ± 50 % of the rated value. In our simulations, we compared the operating conditions where the inductance L was reduced to 0.25L and increased to 4L. Even with long-term parameter drift, it is unlikely to be more severe than the scenarios we have compared in our paper. Under these extreme conditions, our method still maintains good performance. Therefore, our method can theoretically overcome the impact of long-term parameter drift. Please see Fig.12 on Page 15 for details.
Comment 6. Asensitivity analysis on motor inductance, resistance variations, and external load disturbances should add more credibility to the approach.
Response. Thank you for your valuable suggestions on the robustness analysis of the method in this paper. We fully agree that the sensitivity analysis of parameter variations and external disturbances is crucial for verifying the reliability of the control strategy. [R1] performs sensitivity analyses on parameters such as inductance and resistance, but this part of the analysis is omitted in this paper to avoid redundancy due to the small effect of mismatch in the resistance parameter. Therefore, this paper only analyzes the sensitivity of inductance parameters. Sensitivity analysis of external load disturbances by sudden load changes is a relatively common method.

Reviewer 2 Report
Comments and Suggestions for Authors
This article describes three-vector based model for PMSM systems (applied for a robotic system). It focuses on models of the control system, and compares some new methods with conventional methods. The description of the theory with eqautions is useful for any student that is new in the subject of vector control of PMSM.
The introduction is good and gives the context, but some things could be made more clear. Why is the focus on SPMSM? and not to PMSM in general? It would make more sense to cover both surface and internal PMSM, but you could still state that the focus of this specific paper is for SPMSM. Why would it only be intended for SPMSM to use this methods? Is there any specific reason? Is it not functioning for IPMSM, or just not suitable for IPMSM?
In section 2.1, row 95, equation 1 there are expressions for currents. But it looks like something could be missing. Many of the terms have other units (and are not currents), and it looks incorrect (is it an mistake here?). Do you have a reference of any derivation for the equations? It could be good to give some explaination of what this actually is expressing.
Is the method only based on comparison in simulations? The method could be improved by verification through experiments and tests, and not just simulations. But the theory and simulations still looks clear and will still be useful for anyone who would try to build this in an application.
The language is good and understandable, and will be helpful for many new in the subject. Still, there are some small notes on the english language: Check the spelling of the word "email" in the affiliation. And you maybe want to call it "conventional" or "common" three-vector control instead of "traditional" three-vector control (in the abstract), the word traditional could sound informal (similar to section 2).
Round 2
Reviewer 1 Report
Comments and Suggestions for Authors
The authors have improved the manuscript accordingly.